# Perceived Stress and Society-Wide Moral Judgment

**DOI:** 10.3390/ejihpe15060106

**Published:** 2025-06-10

**Authors:** Yi Chen, Junfei Lu, David I. Walker, Wenchao Ma, Andrea L. Glenn, Hyemin Han

**Affiliations:** 1Department of Educational Studies in Psychology, Research Methodology and Counseling, The University of Alabama, Tuscaloosa, AL 35487, USA; ychen195@crimson.ua.edu (Y.C.); jlu27@ua.edu (J.L.); diwalker@ua.edu (D.I.W.); 2Department of Educational Psychology, University of Minnesota, Minneapolis, MN 55455, USA; wma@umn.edu; 3Department of Psychology, The University of Alabama, Tuscaloosa, AL 35487, USA; alglenn1@ua.edu

**Keywords:** perceived stress, moral dilemma judgment, neo-Kohlbergian approach, CNI model

## Abstract

This study examines the relationship between perceived stress and society-wide moral judgment by integrating two influential frameworks: the neo-Kohlbergian approach and the CNI model of utilitarian-deontological decision-making. The neo-Kohlbergian approach to moral judgment proposes three moral schemas: (1) Personal Interest (PI), where decisions are self-focused; (2) Maintaining Norms (MN), which emphasizes adherence to social rules and norms; and (3) Postconventional (PC), where universal ethical principles are prioritized. The CNI model for Utilitarian-Deontological judgment features three psychological processes in decision-making: Sensitivity to Consequence, Sensitivity to Norm, and Inaction Preference. A survey study was conducted to measure perceived stress, neo-Kohlbergian moral judgment (using the behavioral Defining Issues Test [DIT]), and the psychological processes underlying utilitarian-deontological decision-making (CNI). The results indicate that higher perceived stress is linked to greater PI schema endorsement, reduced Norm Sensitivity, and increased Consequence Sensitivity. Furthermore, the PI schema mediated the relationship between perceived stress and Norm Sensitivity. These findings provide insights into how stress shapes moral reasoning and decision-making, with implications for psychological and ethical studies.

## 1. Introduction

### 1.1. The Societal Relevance of Moral Judgment Research and Its Relationship with Perceived Stress

Understanding how individuals make morally significant society-wide decisions is a critical question in modern society. Moral judgments are intertwined with interpersonal relationships ([38]), social policy ([20]), and cultural norms ([10]). For example, during the COVID-19 pandemic, healthcare providers faced ethical challenges in deciding how to allocate scarce resources, such as ventilators ([12]; [65]). Similar moral decisions frequently arise in other contexts, including disaster response and organ transplantation, where choices can impact individual lives and the collective good (e.g., [111]). Given the profound consequences of such moral decisions, it is essential to explore the factors that shape these judgments and understand the psychological processes that underlie them.

Perceived stress plays a critical role in moral decision-making. The impact of stress on decision-making has been of particular interest to researchers ([106]). Stress affects both cognitive and emotional processes, often impairing an individual’s ability to consider multiple perspectives or effectively evaluate long-term consequences (e.g., [22]; [109]). Under stress, individuals may rely more heavily on intuitive or automatic responses, which can skew moral reasoning toward self-serving or immediate outcomes ([128]).

Two primary frameworks are available in moral psychology to study society-wide moral judgment and individual differences: the neo-Kohlbergian approach and sacrificial moral dilemmas involving a balance of utilitarian and deontological decision-making ([86]). These frameworks are valuable for examining how individuals approach complex moral decisions and how various factors, such as perceived stress, interact with their moral reasoning. In the following sections, both frameworks are introduced, and their relationship with perceived stress is explored based on the extant literature.

### 1.2. Neo-Kohlbergian Approach to Moral Judgment

We intend to review the neo-Kohlbergian model of moral functioning as a classical theoretical framework that constitutes the basis for the current study. Although its original development occurred decades ago (e.g., [92]), it has been regarded as one of the mainstream theoretical frameworks in the field of moral psychology and education based on evidence from basic and applied research ([50]). The neo-Kohlbergian approach to moral judgment improves upon Kohlberg’s original theory by replacing stages with schemas. Instead of fixed universal stage-based structures of moral thinking, the neo-Kohlbergian cognitive-developmental framework holds that people progress through “soft” stages, incorporating a shifting emphasis between levels of moral judgment until complex moral judgments become more frequent. Schemas provide specific and tangible representations of moral thinking and address how individuals grasp, order, and emphasize moral content ([93]; [115]). The move away from stages to schemas related to community-wide moral judgment involves three levels of moral judgment: personal interests (PI), maintaining norms (MN), and postconventional (PC) thinking ([93]). Individuals operating within the PI schema prioritize their own interests, judging moral “rightness” based on whether their personal needs are satisfied. In contrast, MN reasoning emphasizes the maintenance of social order and adherence to legal and other norms. Finally, the PC schema, considered the most advanced on the neo-Kohlbergian scale, involves balancing multiple interests to benefit society as a whole.

Another key feature of the neo-Kohlbergian approach is the four-component model comprising an expanded understanding of moral agency, incorporating moral sensitivity, moral motivation, character, and moral judgment ([93]). Moral sensitivity is the ability to recognize that one’s actions or inactions have potential consequences for others and to identify the moral aspects of a given situation. Moral judgment involves determining the most ethically appropriate course of action. Moral motivation refers to the prioritization of moral values over personal interests and reflects a commitment to doing what is morally right. Lastly, moral character is defined as the capacity and determination to carry out the morally optimal course of action despite challenges or obstacles ([92]). Together, these four components influence how individuals perceive, evaluate, and act in moral situations. Notwithstanding the four-component model, neo-Kohlbergians have amassed the most expertise in the moral judgment component, and its measurement is “conceptually the most directly linked to Kohlberg’s original view of moral judgment development” ([115]). 

In recent studies in moral psychology education, we can still see that the neo-Kohlbergian theory has been employed as their theoretical and conceptual basis. For instance, in recent studies on professional ethics education, which were interested in the cultivation of integrative moral functioning embracing moral motivation and behavior, the neo-Kohlbergian model, including both the four-component and schema models, inspired their educational and assessment methodology (e.g., [28]; [27]). Furthermore, recent research on *phronesis* (practical wisdom), which has been suggested as a core capacity for optimal moral functioning and flourishing by virtue ethicists and moral psychologists, has also been informed by the neo-Kohlbergian model ([33]). Their integrative model of wisdom, embracing psychological capacities in the domains of reasoning, emotion, motivation, and behavior, was significantly inspired by the four-component model proposed by the neo-Kohlbergians (e.g., [52]; [120]). Some empirical studies have utilized methodological resources developed based on the neo-Kohlbergian model (e.g., [53]; [62]; [81]). Hence, although the original neo-Kohlbergian theory was developed decades ago, it is possible that the theoretical model still plays a fundamental role in informing research projects in moral psychology and education in recent days.

#### Previous Research on Perceived Stress and the Neo-Kohlbergian Approach to Moral Judgment

To date, no direct studies have investigated the relationship between perceived stress and moral judgment using the neo-Kohlbergian approach. However, existing research offers insights into this endeavor. For example, perceived stress has been shown to impair working memory and cognitive flexibility, thereby reducing individuals’ ability to consider multiple perspectives and make complex moral judgments (e.g., [22]; [109]). As moral reasoning often involves evaluating consequences from multiple perspectives and balancing competing interests—tasks that rely on sophisticated cognitive processing—perceived stress may hinder this process, resulting in a more rigid or less adaptable approach to moral issues. 

Moreover, perceived stress is associated with heightened emotional reactivity, which can bias moral decisions toward immediate, emotionally driven responses rather than reflective, principle-based judgments ([106]). As a result, individuals experiencing sustained stress may exhibit a reduced capacity for higher-level moral reasoning, potentially favoring simpler, more self-focused moral schemas over those requiring nuanced consideration of societal or ethical implications. 

While these theoretical insights are compelling, no empirical studies have examined this connection. The current study aims to address this gap by empirically investigating the impact of perceived stress on moral judgment using the neo-Kohlbergian framework.

### 1.3. Utilitarian-Deontological (U-D) Moral Judgments

Another line of research on society-wide moral judgment focuses on the dichotomy between utilitarian and deontological judgments, often studied using sacrificial moral dilemmas inspired by the trolley dilemma ([43]). These dilemmas involve an inevitable trade-off between avoiding harm to others and achieving the greatest good for the greatest number of people ([41]). Deontology prioritizes actions over consequences ([64]), asserting that certain rights or duties (e.g., the principle of “do not harm”) must be upheld in all situations. This reflects an individual’s sensitivity to moral norms. In contrast, utilitarianism advocates actions that promote the greatest good, such as saving more people. The utilitarian principle holds that a moral action is right if it leads to the greatest benefit for the greatest number of people ([13]), highlighting individuals’ sensitivity to consequences.

#### 1.3.1. Perceived Stress and Utilitarian-Deontological Moral Judgment

Research on the effects of stress on decision-making has found that stress tends to shift individuals’ cognitive processes from analytical to more intuitive and habitual reasoning ([128]). This shift aligns with deontological moral inclinations, which rely more on automatic, rule-based responses than deliberative analysis ([46]). Deontological moral inclination is characterized by decisions based on the intrinsic nature of an action, regardless of the consequences ([46]). Indeed, this positive relationship between perceived stress and deontological inclination has been supported by prior studies ([119]; [130]). Similarly, studies on acute stress induction have also found a tendency toward deontological judgments, reinforcing the link between stress and rule-based moral reasoning ([72]; [102]; [107]; [127]).

#### 1.3.2. Measuring Utilitarian-Deontological Moral Judgments

A set of sacrificial moral dilemmas designed by [48] ([48]) is available in moral psychology for researching U-D moral judgment ([41]). Despite their extensive application and contribution to the field, dilemmas have been criticized for their theoretical underpinnings and methodological design. 

Theoretically, [32] ([32]) argued that utilitarian and deontological judgments should not be treated as opposite ends of a single dimension in psychological studies. Instead, they represent distinct and independent psychological processes ([32]). Methodologically, the traditional format of sacrificial dilemmas does not experimentally manipulate utilitarian or deontological judgments ([42]). Instead, all dilemmas employ a proscriptive moral norm, where the action should not be taken based on a moral norm, and the benefits of the action always outweigh the costs. This design limits the ability to isolate distinct cognitive and moral processes.

To address these limitations, [41] ([41]) developed the CNI model, which was designed to disentangle three dimensions in sacrificial moral dilemma judgment: Sensitive to Consequence (C, corresponding to the theoretical interpretation of utilitarian judgment, focusing on the benefits and costs of an action), Sensitive to Norms (N, corresponding to the theoretical interpretation of deontological judgment, emphasizing adherence to moral rules), and general Inaction Preference (I). The final dimension accounts for the human tendency to prefer inaction over action in the face of uncertainty ([118]). One example of this tendency is that harm caused by action is often perceived as more severe than harm caused by inaction (e.g., pushing someone to their death is perceived as more severe than witnessing someone die without intervening; [105]). The inclusion of Inaction Preference represents a significant theoretical advancement, addressing the inherent human bias toward inaction when faced with moral dilemmas.

Since its introduction, the CNI model has been widely adopted in diverse research contexts, including studies on moral dilemma judgments and psychopathy ([66]), incidental emotions ([43]), foreign language effects ([15]), and perceived stress ([130]), political ideology ([74]), and personality traits ([68]), among others.

Despite its substantial theoretical contributions, the CNI model has a notable methodological limitation: its inability to generate individual-level scores ([41]). This drawback restricts its use to between-subjects designs and makes it unsuitable for correlational studies, significantly limiting its broader application. To overcome this limitation, [24] ([24]) proposed an EIRTree-CNI model that adopted an extended item response tree (EIRTree) model on the CNI test. The EIRTree model allows for the direct estimation of individual scores. It has been applied to self-reported decision-making measures (e.g., [18]) and strategies using educational assessments (e.g., [77]). The current study adopts the CNI test to measure U-D moral judgments and uses the EIRTree-CNI model for the data analysis.

[130] ([130]) employed the CNI model to examine the psychological mechanisms underlying U-D moral judgments in the context of perceived stress. They found that the primary difference between the high-stress and low-stress groups was an increased preference for inaction among those experiencing higher stress. The groups did not differ significantly in their sensitivity to consequences or sensitivity to norms.

At first glance, this finding does not contradict the results of studies using traditional moral dilemmas, where questions employ only proscriptive norms and the benefit of action outweighs the cost. In this type of moral dilemma, the deontological inclination encompasses the tendency toward inaction ([41])—the response is “no” regardless of whether the decision was driven by sensitivity to a moral norm or a preference for inaction. [130] ([130]) further supported this by conducting supplemental analysis on a subset of dilemmas aligned with traditional designs (i.e., proscriptive norms, action benefits exceeding costs). The results confirmed that the high-stress group exhibited stronger deontological or weaker utilitarian inclinations than the low-stress group, using the traditional measurement standard.

Given that [130]’s ([130]) study is the only one to date exploring the relationship between perceived stress and moral judgment using the CNI model, replication is necessary ([83]), which the current study seeks to address. The necessity of this replication is further underscored by the fact that [130] ([130]) recruited participants in China, meaning that no study on this topic has yet been conducted with an American sample.

Additionally, [130]’s ([130]) sample may have introduced biases related to action preference and stress level. The participants in their study exhibited a general preference for action, which deviated from the typical inaction preference observed in moral judgment research. This discrepancy challenges one of the foundational assumptions of the CNI model. Furthermore, the average stress level reported in their sample was notably high (*M* = 26.51). The coexistence of high-stress levels and an atypical preference for action creates a paradox, as these traits seemingly contradict the conclusion that higher stress leads to greater inaction. This inconsistency further highlights the necessity of replicating this study with a different sample.

### 1.4. Research Goals and Hypotheses

The current study incorporates two widely adopted frameworks in moral judgment research, the CNI model and the neo-Kohlbergian approach, to explore the relationship between perceived stress and moral judgment. The CNI model examines individuals’ U-D tendencies and dissects moral decision-making into three distinct components—Sensitivity to Consequence, Sensitivity to Norm, and Inaction Preference. The neo-Kohlbergian approach focuses on developmental moral reasoning through cognitive schemas, offering insights into the progression of moral thinking. By integrating these two frameworks, this study aims to provide a comprehensive understanding of how perceived stress interacts with society-wide moral judgment.

Furthermore, this study examined whether moral reasoning schema adoption mediates the relationship between perceived stress and the CNI model dimensions. [23] ([23]) has found a significant correlation between schema adoption and the CNI dimensions. This integration is particularly relevant because of the complementary focus of the frameworks: the CNI model emphasizes decision outcomes, while the neo-Kohlbergian approach explores the cognitive reasoning processes behind those outcomes. Examining moral reasoning schemas as mediators highlights the nuanced interplay between thought processes and decision-making tendencies, offering a dynamic perspective on the effects of perceived stress.

#### 1.4.1. Perceived Stress and Utilitarian-Deontological Moral Judgments

Although [130]’s ([130]) methodology may be subject to scrutiny (see Section 1.3.2), their primary finding—individuals with higher levels of perceived stress exhibited a greater preference for inaction—is plausible. From an embodied cognition perspective, excessive stress can trigger a freeze response, which is a survival mechanism observed in animals ([69]). Similarly, stress has been negatively associated with physical activity ([110]), suggesting a general reduction in action-oriented behaviors. This evidence supports the hypothesis that individuals experiencing higher stress levels are more inclined toward inaction when confronted with moral dilemmas.

Regarding Sensitivity to Consequence (C) and Sensitivity to Norm (N), the findings are not consistent across studies. [130] ([130]) did not find any relationship with perceived stress. [68] ([68]) manipulated time pressure by instructing half of the participants to complete moral dilemmas under strict time constraints and found that the time-pressured group scored lower on the Consequence Sensitivity. [72] ([72]), using the Trier Social Stress Test to induce acute stress, observed that participants under acute stress displayed higher Norm Sensitivity and Inaction Preference, with no significant changes in Consequence Sensitivity. Although these studies yielded inconsistent conclusions, they suggest that stress may bias individuals toward deontological reasoning—characterized by increased Norm Sensitivity or reduced Consequence Sensitivity.

It is worth noting that findings on acute stress may not fully generalize to chronic or perceived stress, as the cognitive effects of acute stress align with those of chronic stress only when the acute stress levels are extremely high ([97]). Nevertheless, studies on induced acute stress have shown a consistent pattern with those using traditional moral dilemmas, suggesting that higher stress levels are associated with less utilitarian and more deontological judgments (e.g., [107]; [127]). Given that stress has consistently been shown to impair deliberative reasoning and shift individuals toward intuitive, emotion-driven decision-making ([128]), and that utilitarian judgments rely heavily on effortful cognitive processing ([48]), higher perceived stress levels are expected to correlate with increased Norm Sensitivity or decreased Consequence Sensitivity.

This expectation also aligns with dual-process theory, which proposes that moral decisions result from the interplay between intuitive, emotion-driven processes (System 1) and deliberative, reasoning-based processes (System 2). Under stress, the capacity for deliberative reasoning (System 2) is often compromised, leading individuals to rely more heavily on intuitive processes (System 1). Consequently, higher perceived stress levels are anticipated to correlate with increased Norm Sensitivity or decreased Consequence Sensitivity, reflecting a shift toward deontological reasoning.

In summary, the current study hypothesizes that participants with higher stress levels demonstrate a stronger Inaction Preference, along with increased Norm Sensitivity and/or decreased Consequence Sensitivity.

#### 1.4.2. Perceived Stress and the Neo-Kohlbergian Approach to Moral Judgment

To date, no direct studies have investigated the relationship between perceived stress and moral judgment within the Neo-Kohlbergian framework. However, indirect evidence suggests a potential association. For instance, [101] ([101]) demonstrated that a mindfulness-based stress reduction intervention significantly enhanced participants’ moral reasoning abilities, as evidenced by increased postconventional schema adoption. This finding suggests that perceived stress may negatively correlate with higher-order moral reasoning, as stress reduction is associated with improved reasoning. 

Higher-level moral reasoning, such as the adoption of the PC schema, requires the ability to weigh competing interests, evaluate multiple perspectives, and make nuanced ethical judgments—all of which rely on complex cognitive functions. However, perceived stress impairs cognitive flexibility and working memory (e.g., [22]; [109]), both of which are crucial for moral reasoning. This impairment may lead to a narrower and less flexible approach to moral issues. Additionally, perceived stress heightens emotional reactivity, which may bias moral decision-making toward immediate, emotionally driven responses rather than reflective, principle-based judgments ([106]). Consequently, individuals experiencing higher levels of perceived stress may rely more on simpler, self-focused moral schemas rather than those requiring nuanced consideration of societal or ethical implications.

#### 1.4.3. Sex as a Controlling Variable

Sex differences in moral judgment and stress perception necessitated controlling for sex in this study. A meta-analysis by [40] ([40]), covering 40 studies with over 6000 participants, found that men tend to make more utilitarian decisions than women do in moral dilemmas. This finding was also supported by an international sample ([6]). Adopting the CNI model, [41] ([41]) provided further insight into this sex difference, showing that women exhibited higher Norm Sensitivity and Inaction Preference than men. In their replicated study, [41] ([41]) also found that women scored higher in Consequence Sensitivity than men. 

Sex differences have also been examined within the neo-Kohlbergian framework. A meta-analysis by [114] ([114]), covering 56 samples (*n* > 6000), found that women scored significantly higher than men in moral reasoning, although the effect size was small. For instance, among college students, the average PC score was 44.11 (*SD* = 12.21, *n* = 449) for men and 45.88 (SD = 12.19, *n* = 436) for women. A more recent study by [100] ([100]), focusing on accounting students, found a greater sex disparity in moral reasoning scores: the average P-score was 25.89 (*SD* = 12.01, *n* = 56) for men and 31.09 (SD = 15.05, *n* = 160) for women. The significant sex difference favoring women was also supported by two additional meta-analyses ([49]).

Gender plays a crucial role not only in moral judgment but also in self-reported stress. Women consistently report higher perceived stress levels than men in general adult populations ([4]) and college student samples ([45]). Female students, in particular, report greater academic stress, including exam-related pressures ([112]) and general academic demands ([90]). 

Given that the current sample is relatively homogeneous apart from sex and that women are disproportionately represented controlling for sex is critical to avoid biased results. Therefore, sex was included as a control variable in all analyses to ensure the robustness of the findings.

## 2. Method

This study employed a cross-sectional research design to investigate how perceived stress, measured using the Perceived Stress Scale-10 (PSS; [31]), influences moral judgment. Moral judgment is assessed using two frameworks: the neo-Kohlbergian approach, measured by the behavioral Defining Issues Test (bDIT), and the CNI model, which evaluates responses to U-D sacrificial moral dilemmas.

### 2.1. Procedure

The study was conducted at a university in the southeastern United States using a student research participation platform (Sona system). The Institutional Review Board (IRB) of the university reviewed and approved this study. Through the Sona system, students could browse a range of available research projects and voluntarily select those that interested them. Participants who opted for the current study began by reading an informed consent form on the survey’s interface. They were informed of their right to withdraw from the study at any time without penalty. The survey was open for increments between February 2021 and August 2022.

After consenting to participate, the students completed the study questionnaire, which was presented in the following order: CNI model items ([41]; [66]), bDIT ([56]), Perceived Stress Scale ([31]), and demographic questions.

### 2.2. Participants

A total of 618 college students participated in the study, of whom 588 completed all the questions. To ensure high-quality responses, the data were screened using attention-check items, survey completion time, and the *Careless* package in *R* to identify and remove low-quality responses.

Responses were excluded if participants failed the attention-check item or completed the survey in less than 10 min. The completion time threshold was based on [91]’s ([91]) finding that college-educated adults can read at an average rate of 400 words per minute. The current survey contained approximately 4000 words (excluding consent forms, repeated question stems, and unrelated demographic options), which required at least 10 min for a typical college student to complete. After applying these criteria, data from 477 participants remained.

Additionally, the careless package created by [121] ([121]) was used to identify and exclude low-quality responses. Entries with an unusually long string of identical responses were flagged and removed. A cutoff point of 0.4 standard deviations above the mean was applied following [126] ([126]). This process resulted in a final dataset of 337 participants, which closely aligns with the sample sizes of other studies utilizing the CNI model (e.g., [15]; [75]; [76]).

An a priori power analysis was conducted using G*Power version 3.1.9.7 ([39]) to determine the minimum sample size required to test the study’s hypothesis. The results indicated that the required sample size to achieve 80% power for detecting a medium effect (effect size of 0.3 for Pearson Correlation and 0.15 for multiple linear regression, criteria based on [29]), at a significance criterion of *α* = 0.05, was 84 for Pearson correlation test, and 68 for multiple linear regression with two predictors. Thus, the final sample size of 337 was adequate to test the study hypothesis.

Among the 337 college students in the final dataset, 285 (84.6%) were female, and 49 (14.5%) were male. Participants’ ages ranged from 18 to 57 years, with a mean age of 22 years and a standard deviation of 6.2. Regarding race/ethnicity, 286 (84.9%) identified as Caucasian, 32 (9.5%) as African American, 1 (0.3%) as Asian, 4 (1.2%) as “other”, and 14 (4.2%) as multiracial. Academic classification included 22 (6.5%) first-year students, 172 (51%) sophomores, 66 (19.6%) juniors, 47 (13.9%) seniors, and 30 (8.9%) students who reported being beyond their fourth year in college.

### 2.3. Materials

#### 2.3.1. CNI Model

The 24-item CNI test includes four types of items that are the products of two dimensions: Norm (Proscriptive or Prescriptive) and Consequence (Larger or Smaller benefits compared to costs) ([41]; [66]). It was designed to measure participants’ Sensitivity to Norms, Sensitivity to Consequences, and Inaction/Action Preference. To analyze the data and generate individual-level CNI scores, the EIRTree-CNI Model will be employed ([24]). 

[66] ([66]) expanded upon [41]’s ([41]) original CNI test by adding 24 new moral dilemmas. The current study adopts these additional dilemmas for several reasons, as follows.

[9] ([9]) highlighted a potential issue with the original CNI model, suggesting that participants might interpret dilemmas differently from researchers regarding whether the benefits outweigh the costs or which norm should be prioritized in a given scenario. To address this, Gawronski and colleagues reanalyzed data from eight sub-studies in their original 2017 research to assess the validity of the consequence and norm operations within their model ([44]). Specifically, they compared “action” responses between items with proscriptive versus prescriptive norms and between items where benefits were greater or smaller than costs. Their findings indicated that all six original scenarios were generally valid, except for the Abduction Dilemma, which showed no significant norm effect across all eight sub-studies and was thus excluded from the current study. However, even among the other five scenarios, at least one sub-study for each scenario showed non-significant effects for either the norms or consequences.

Using the same method, we tested the norm and consequence effects for the six new scenarios introduced by [66] ([66]), reanalyzing the data from two sub-studies in their research. The results showed that all norm and consequence effects were statistically significant (see Table 1 for a detailed analysis). This validation provides confidence in selecting the six new scenarios for the current study instead of those used in [41] ([41]).

Furthermore, the six original scenarios in [41] ([41]) are relatively well-known and may have been encountered by participants in their everyday lives or discussed with others. As a result, participants’ decisions may reflect social expectations rather than genuine moral intuitions. Additionally, as this study was conducted during the COVID-19 pandemic, participants might have been particularly sensitive to two of [41]’s ([41]) scenarios—the Immune Deficiency Dilemma and the Vaccine Dilemma. This heightened sensitivity could lead to overthinking or responses influenced by the pandemic context rather than the participants’ typical moral reasoning. To mitigate these concerns and ensure that the study captured intuitive moral judgments, the 24 newly developed moral dilemmas introduced by [66] ([66]) were chosen for the current study.

#### 2.3.2. The Behavioral Defining Issue Test

The behavioral Defining Issue Test (bDIT; [56]) is the most recently developed questionnaire for measuring moral reasoning development, built on the widely used DIT-1 ([93]; [113]). The bDIT consists of three moral stories that center on conflicts primarily related to social rights and legal issues. After each story, the participants were asked to make a decision regarding the scenario and then indicate their major concerns by answering eight multiple-choice questions. Each question provides three response options corresponding to the Personal Interest, Maintaining Norm, and Postconventional schemas.

Three parameters—PI, MN, and PC—were calculated from the participants’ responses, reflecting the proportion of schema-based reasoning applied. The validity and reliability of the bDIT are well supported by previous research ([25]; [51]). Cronbach’s alpha values reported in the original study were 0.79 ([56]), 0.74 ([25]), and 0.77 ([55]), demonstrating acceptable internal consistency. [51] ([51]) also conducted measurement invariance tests and differential item functioning tests, confirming that the bDIT consistently measures the construct of interest across sex, political affiliation, and religious affiliation without bias in its items. In the present study, Cronbach’s alpha was 0.76, indicating reliable internal validity.

#### 2.3.3. The Perceived Stress Scale-10

The Perceived Stress Scale-10 (PSS; [31]) is a widely used instrument for measuring perceived stress. It comprises ten items that assess how often participants experience certain feelings over the past month. An example item is “In the last month, how often have you felt that you were unable to control the important things in your life?”. Respondents rated the questions on a 5-point Likert scale (0 = never, 4 = very often). Four items are reverse-scored (e.g., “In the last month, how often have you felt that things were going your way?”), and the scores for all items were summed to produce a final stress score.

The PSS is designed for easy comprehension and requires only a junior high school reading level. Reliability is robust, with Cronbach’s alpha ranging from 0.74 to 0.91 across various studies ([70]; [103]). Its validity has been well-documented across diverse populations and time periods ([16]; [70]; [87]). In the current study, Cronbach’s alpha was 0.88, indicating excellent reliability.

#### 2.3.4. Attention Check

To ensure data quality, an attention-check question was included within the PSS scale, framed as “In this item, we want to have an attention check. Please click Very Often here”. Only responses following the instructions were considered valid.

### 2.4. Data Analysis

All statistical analyses were performed using SPSS 24.

#### 2.4.1. Correlation Analysis

Pearson’s correlation analyses were performed to examine the relationships among the key variables: perceived stress, moral reasoning schemas, and dimensions of the CNI model. Statistical significance was defined as a *p*-value ≤ 0.05. [29]’s ([29]) guidelines were used to interpret effect sizes, with values of 0.10 indicating small effects, 0.30 moderate effects, and 0.50 large effects. 

#### 2.4.2. Main Effect While Controlling for Sex

Multiple linear regressions were conducted to assess the relationship between perceived stress and moral judgment while accounting for sex differences. Participants who identified as male or female were included due to insufficient representation in other sex categories. Additionally, independent *t*-tests were performed for each of the six variables in the bDIT and CNI models to determine whether there were any sex differences in these variables independently, without controlling for perceived stress.

Assumptions for multiple linear regression were thoroughly tested: Mahalanobis Distances ([71]) were calculated to detect multivariate outliers. [8] ([8]) indicated that a value above 15 is problematic for a sample smaller than 500 with fewer than five predictors. The Durbin-Watson test ([58]) was conducted to check whether the residuals were independent. A value close to 2 is considered normal and indicates no significant evidence of serial correlation. The histogram of the standardized residuals and Normal P-P Plot were plotted to check if the residuals were normally distributed ([98]). An almost symmetrical distribution of the histogram or lying around the diagonal line in the P-P Plot suggests no violation of the normality assumption. Homoscedasticity ([125]) was tested by plotting studentized residuals against unstandardized predicted values. An almost equally distributed scatter plot supports homoscedasticity. To check for multicollinearity, variance inflation factors (VIFs) were computed for the dependent variables in both the bDIT and CNI models. A VIF value below 4 was expected, indicating the absence of multicollinearity ([17]).

#### 2.4.3. Exploring Quadratic Relationship

Quadratic regression analyses were performed to investigate potential nonlinear relationships. Significant quadratic relationships were further examined by comparing model fits with linear models using the Likelihood Ratio (LR) test, Akaike Information Criterion (AIC), and Bayesian Information Criterion (BIC; [21]). If the quadratic term significantly improved the model fit (*p* < 0.05 for the LR test and lower AIC/BIC values), the quadratic model was preferred. Otherwise, a simpler linear model was retained. The turning point in curvilinear relationships was calculated using the formula $x=-\frac{\beta 1⧈}{2\beta 2}$.

#### 2.4.4. The Mediation Role of Moral Reasoning Development

Building on [23]’s ([23]) findings that moral reasoning schema adoption is correlated with decision-making in the CNI model, this study explored whether these schemas mediate the relationship between perceived stress and the CNI model. Since a significant mediation effect requires a significant correlation between the independent variable and the mediator, only bDIT variables significantly correlated with PSS were included as mediators.

Mediation analyses were conducted using the PROCESS macro for SPSS ([60]), specifically Model 4, which tests simple mediation. This approach estimates the direct, indirect, and total effects of the independent variable (IV) on the dependent variable (DV) through a mediator (M).

Macro employs bootstrapping procedures to generate confidence intervals (CIs) for indirect effects, which provides a robust test of mediation. In this study, bootstrapping procedures (5000 resamples) with 95% bias-corrected CIs were employed to assess the significance of indirect effects. Mediation was considered significant if all indirect and total effects were significant and the CI for the indirect effect did not include zero.

## 3. Results

The summary index scores and the correlation statistics are presented in Table 2. 

Compared to [24] ([24]), who utilized a published dataset collected by [66] ([66]; https://osf.io/ndf4w/, accessed on 6 April 2025), the current study yielded a lower C score (−0.80 compared to −0.63 in Study 1), a similar N score (both 0.40), and a higher I score (0.40 compared to 0.23 in Study 1). Independent *t*-tests revealed significant differences between the C scores (*t* = 2.752, *p* = 0.006) and I scores (*t* = −3.071, *p* = 0.001) but not the N scores (*t* = 0.003, *p* = 0.998).

In terms of the bDIT scores generated in the current study, while only the PC score has been widely reported in the existing literature, it is comparable to those reported in previous studies. The mean PC scores were 52.24 ([54]), 51.23 ([56]), 49.82 and 52.51 from two studies by [56] ([56]), and 49.15 ([25]). The mean PC score in the present study was 52.51, which closely aligns with these findings. Among all participants, 219 predominantly adopted the PC schema, 53 the MN schema, and 39 the PI schema. These results align with the prior literature, indicating that young adults and college students predominantly employ the PC schema in their moral reasoning ([93]).

Regarding PSS, the mean score of 20.91 observed in the current study was significantly higher than the norm sample average of 13.06 ([31]). However, this is consistent with recent studies reporting mean scores of 23.89 ([89]) and 19.00–21.92 ([85]).

### 3.1. Perceived Stress and Neo-Kohlbergian Approach to Moral Judgment

Pearson correlational analyses (See Table 2) found a significantly positive correlation between PSS and PI schema adoption (*r*_(335)_ = 0.11, *p* = 0.049), suggesting that participants under higher perceived stress are more likely to prioritize the PI schema compared to those who experiencing lower stress. No significant correlations were found between the PSS and MN or PC schemas.

Multiple linear regression analyses were conducted with PSS and sex as independent variables and bDIT schema variables as dependent variables. All the assumptions were satisfied. A model with a *p*-value slightly higher than 0.05 was found for PI (*F*_(2, 331)_ = 2.59, *p* = 0.077), where the *p*-value of the effect of perceived stress was similar (*t*_(332)_ = 1.94, *p* = 0.054). Sex had no significant effect on PI schema adoption. Together, perceived stress and sex accounted for 1.5% of the variance in the PI schema scores. No significant regression models were observed for the MN or PC schemas.

Independent *t*-tests comparing sex differences across the three moral reasoning schemas yielded no significant results (p values for each of the schemas: PI: *p* = 0.236, MN: *p* = 0.655, PC: *p* = 0.579), suggesting that sex does not play a significant role in moral reasoning schema adoption.

A significant curvilinear relationship was found between the PSS and PI schema (*Adjusted R*^2^ = 0.013, *F*_(2,334)_ = 3.14, *p* = 0.045). However, the quadratic model did not significantly improve the model fit compared to the linear model (*F*_(1, 334)_ = 2.35, *p* = 0.127).

### 3.2. Perceived Stress and the CNI Model Materials

Pearson correlational analyses (Table 2) found that PSS score was significantly and negatively correlated with Norm Sensitivity (*r*_(335)_ = −0.19, *p* = 0.001) and positively correlated with Consequence Sensitivity (*r*_(335)_ = 0.12, *p* = 0.03). These findings suggest that participants experiencing higher levels of perceived stress were more sensitive to the consequences and less sensitive to the norms when making judgments in sacrificial moral dilemmas.

Multiple linear regressions were conducted with PSS and sex as independent variables and CNI model dimensions as dependent variables. All assumptions for the regression analysis were met. Significant models were found for Consequence Sensitivity (*F*_(2, 331)_ = 6.79, *p* = 0.001) and Norm Sensitivity (*F*_(2, 331)_ = 8.06, *p* < 0.001), but not for Inaction Preference.

For Consequence Sensitivity, both stress (*t*_(332)_ = 2.28, *p* = 0.023) and sex (*t*_(332)_ = −2.93, *p* = 0.004) were significant predictors, together accounting for 3.9% of the variance in Consequence Sensitivity. For Norm Sensitivity, stress (*t*_(332)_ = −3.62, *p* < 0.001) had a significant effect, while the effect of sex was marginally significant (*t*_(332)_ = 1.79, *p* = 0.075). Stress and sex together accounted for 4.6% of the variance in Norm Sensitivity.

Independent *t*-tests comparing sex differences across the CNI model dimensions revealed significant differences in Consequence Sensitivity (*t*_(2, 332)_ = 2.881, *p* = 0.004), as well as marginally significant differences in Norm Sensitivity (*t*_(2, 332)_ = 1.704, *p* = 0.089) and Inaction Preference (*t*_(2, 332)_ = 1.752, *p* = 0.081).

Sex differences in the CNI model dimensions showed that compared to males (C: *M* = −0.57, *SD* = 0.69; N: *M* = 0.25, *SD* = 0.71; I: *M* = 0.27, *SD* = 0.59), females (C: *M* = −0.83, *SD* = 0.57; N: *M* = 0.43, *SD* = 0.67; I: *M* = 0.42, *SD* = 0.55) exhibited lower Consequence Sensitivity, higher Norm Sensitivity, and higher Inaction Preference.

A significant curvilinear relationship was also found between the PSS and Norm Sensitivity (*adjusted R*^2^ = 0.034, *F*_(2, 334)_ = 6.87, *p* = 0.001). However, the quadratic model did not significantly improve the model fit (*F*_(1, 334)_ = 1.46, *p* = 0.227), indicating that the linear relationship provided a more parsimonious explanation.

### 3.3. The Mediation Effect of Moral Reasoning Schema on Perceived Stress and CNI Model

Based on the results of the correlational analyses, the only variable in the bDIT model that was significantly correlated with the PSS was the PI schema. Consequently, three mediation models were tested with PI schema adoption as the mediator between PSS and the three dimensions of the CNI model.

The mediation analysis revealed significant findings for the relationship between PSS and Norm Sensitivity (Figure 1). PSS significantly predicted PI schema adoption (Path a, *b* = 0.003, *SE* = 0.001, *p* = 0.049). In turn, PI schema adoption significantly predicted Norm Sensitivity, controlling for PSS (Path b, *b* = −0.89, *SE* = 0.22, *p* = 0.001). The total effect of PSS on the N index was significant (Path c, *b* = −0.018, *SE* = 0.005, *p* = 0.001). The direct effect of PSS on N was significant (Path c’, *b* = −0.016, *SE* = 0.005, *p* = 0.002), suggesting a partial mediation.

The indirect effect of PSS on Norm Sensitivity through PI schema adoption (Path a × b) was also significant (*b* = −0.002, *SE* = 0.001), with a 95% bias-corrected confidence interval of −0.0048 to −0.0001, which did not include zero, confirming the mediation effect. 

Including the mediator improved the model fit, although small, increasing the explained variance in the outcome variable from *R*^2^ = 0.0353 to *R*^2^ = 0.0793. This represents an *R*^2^ change of 0.044, indicating that PI accounted for an additional 4.4% of the variance in Norm Sensitivity beyond the effect of perceived stress alone.

However, the mediation models examining the effects of PSS on Consequence Sensitivity (C) and Inaction Preference (I) through PI schema adoption were not supported.

## 4. Discussion

This study examined the relationship between perceived stress and moral judgment using two theoretical models: the neo-Kohlbergian approach and the CNI model. The findings suggest that perceived stress influences moral judgment differently within each framework. Specifically, higher stress levels were associated with a higher likelihood of individuals adopting the PI schema, increased Sensitive to Consequences, and reduced Sensitive to Norms. These findings remained significant even after controlling for sex. Additionally, the PI schema was found to mediate the relationship between perceived stress and the CNI model, although the size was small. The following sections explore these findings in more detail.

### 4.1. Findings About Neo-Kohlbergian Approach to Moral Judgment

#### 4.1.1. The Positive Correlation Between Perceived Stress and PI Schema

Among the three moral reasoning schemas, only the PI schema was positively correlated with perceived stress, as measured by the PSS. This relationship remained marginally significant even after controlling for sex. This finding supports the hypothesis that higher stress levels are associated with simpler, more self-focused moral schemas rather than principle-based judgments. 

Evidence from prior studies aligns with this interpretation. Stress has been linked to egoistic decision-making and self-serving motivations in moral contexts, as shown in studies on acute stress ([104]), perceived stress ([119]), and physiological indicators of stress, such as cortisol levels ([108]). Although some contradictory results have also been discovered—for example, acute stress led to altruistic moral choice ([102]) or altruistic behavior ([104])— these have been attributed to individuals attempting to alleviate their own distress or the cognitive burden of aversive emotions. For example, stress significantly increases the frequency of donations but decreases the amount donated ([104]). Donating has been found to promote happiness ([1]) and alleviate negative mood ([26]). Decisions about whether to donate are often driven by the motive of enhancing one’s mood, while decisions regarding the amount donated are influenced more by empathic concern for the recipient ([35]). These findings suggest that stress consistently pushes individuals toward self-focused behavior, although the specific manifestation varies depending on the context and underlying motivations. 

Perceived stress may also trigger cognitive or emotional regression, causing individuals to revert to less-advanced moral reasoning schemas. Research suggests that stress can lead to a reliance on earlier learned behavioral patterns ([11]). [2] ([2]) found that stress caused participants to adopt “childlike” behaviors, such as favoring in-group benefits even when unfair, reducing sensitivity to contradictions, and generally becoming more self-serving. However, it remains unclear whether this is a result of stress-induced deterioration in social interaction ([36]), intellectual regression ([99]), or a combination of both. A longitudinal qualitative study could provide valuable insights into this issue.

#### 4.1.2. Perceived Stress and MN/PC Schema

Contrary to the hypothesis, higher perceived stress levels were not negatively correlated with principle-based judgments (PC schema). One possible explanation is that the observed increase in PI schema adoption reduces MN and PC schemas by approximately equal amounts, thereby diluting their measurable effect. Given that the sum of the three schema adoptions must equal 100%, stress-driven shifts toward PI inevitably decrease MN and/or PC. A linear regression between the three schemas (IV: MN and PC; DV: PI) found that the changes in MN and PC were relatively evenly distributed along PI’s change, |*β*_MN_| (1.129) ≈ |*β*_PI_| (1.227). Given that the correlation coefficient between stress and PI was already small (0.11), the effect on MN or PC may not have been detectable.

Interestingly, [94] ([94]) identified a curvilinear relationship between anxiety and MN schema adoption, where moderate anxiety levels were associated with heightened prioritization of the MN schema. The discrepancy between [94] ([94]) and the present findings may reflect differences in the psychological constructs of perceived stress and anxiety. Moderate anxiety may heighten vigilance and adherence to societal norms as coping mechanisms to reduce uncertainty ([94]). In contrast, perceived stress may deplete the cognitive and emotional resources required for structured moral reasoning. However, the effect size in Roberts’s study was small (*r*^2^ = 0.03), suggesting that the relationship between anxiety and MN schemas may not be particularly robust or impactful.

#### 4.1.3. Sex Differences

Finally, no sex differences were found in the bDIT scores.

Considering the broader trend of findings showing that women tend to score higher on the PC schema, one possible explanation for the lack of sex differences in the current findings could be societal change. Research has found a reduction in gender role differences in modern society ([7]), which may contribute to diminishing disparities in moral reasoning across sexes. Nevertheless, this inference remains theoretical and requires empirical validation through further research.

### 4.2. Findings About the CNI Model

Perceived stress was positively associated with Consequence Sensitivity and negatively associated with Norm Sensitivity. These findings contradict the original hypothesis and deviate from those of previous studies on this topic. Below, I compare these results with those of earlier research and propose potential explanations.

#### 4.2.1. Previous Studies on the Relationship Between Perceived Stress and Utilitarian-Deontological Moral Dilemma Judgment

Three studies have investigated the connection between perceived stress and U-D moral dilemma judgment. [119] ([119]) found that participants under stress were more likely to make deontological decisions compared to those not under stress. [130] ([130]) also reported that higher self-reported perceived stress was linked to more deontological decisions in traditional sacrificial dilemmas. Using the CNI model, [130] ([130]) attributed this association primarily to increased Inaction Preference, similar to the “freeze response” observed in animals under stress ([69]). The discrepancies between these studies and the present study may be due to differences in the sample characteristics. [119] ([119]) studied Lithuanian employees, while [130] ([130]) focused on Chinese college students. These populations likely differ from the American college students in this study in terms of stress perception, life experiences and moral reasoning tendencies.

[67] ([67]) found that the effect of cortisol levels on U-D tendencies was moderated by the motivation to avoid uncertainty. Individuals with low motivation were more likely to exhibit deontological tendencies under stress, aligning with most prior studies. Conversely, for those with high motivation to avoid uncertainty, cortisol levels were associated with a utilitarian tendency, a result consistent with the current study. However, their study included only male participants, whereas the current study predominantly included female participants. While sex differences in the motivation to avoid uncertainty or preserve certainty have not been established ([122]), replicating [67]’s ([67]) study with female participants and self-reported stress measures could offer further insights.

#### 4.2.2. Stress-Related Variables Lead to More Utilitarian Judgments

A seemingly contradictory finding to the widely observed positive correlation between stress and deontological decision-making comes from [131] ([131]), who studied the impact of mental fatigue during the COVID-19 pandemic on deontological decision-making. They found that individuals experiencing higher fatigue with COVID-19-related information were more likely to make utilitarian choices in COVID-19-themed dilemmas. Similarly, [79] ([79]) reported that frontline healthcare workers with higher self-rated stress exhibited a stronger inclination toward utilitarian judgments, although the direct link between stress and moral decision-making was not explicitly tested in their study.

[131] ([131]) intentionally chose impersonal dilemmas, which differ from personal moral dilemmas in that the harm to the victim is caused indirectly, such as through technology, mechanisms, or medicine, rather than through direct physical contact ([84]). Consequently, impersonal dilemmas evoke weaker emotional responses ([47]) and are less likely to pull participants toward a deontological decision ([59]) than personal dilemmas. Studies supporting dual-process theory have mostly adopted personal moral dilemmas or have found significant differences only with personal dilemmas but not impersonal ones ([131]). Of the three studies on non-acute stress and sacrificial moral dilemma judgment to date, two ([67]; [119]) adopted only personal dilemmas.

Mental fatigue accumulates during prolonged stressful situations ([129]). Based on the dual-process theory ([47]), mental fatigue reduces emotional responsiveness ([117]) and diminishes the motivation for deontological decisions. [131] ([131]) expanded this explanation by adopting a conflict model ([14]; [95]). Studies based on dual-process theory ([47]) often focus on the responder’s moral motivation; the conflict model also emphasizes the directive nature of the dilemma itself[note 1]. For example, as mentioned earlier, impersonal items are less likely to elicit deontological decisions because they do not strongly engage participants’ emotional responses. Taken together, in dilemmas where emotional engagement is limited—such as impersonal dilemmas—individuals experiencing higher fatigue may be more inclined to rely on utilitarian reasoning.

[131]’s ([131]) theory does not fully account for the current study’s findings because the dilemmas used in this study included a mix of personal and impersonal dilemmas. Nevertheless, one possible explanation is that individuals under higher stress may develop emotional callousness or desensitization toward harming others, freeing them from the impulse to make deontological decisions. Supporting this notion, [78] ([78]) found that while acute stress led to fewer utilitarian decisions, participants with high levels of Machiavellianism and narcissism were more likely to make utilitarian choices than those with lower levels of dark personality traits. Certain types of narcissism and Machiavellianism are strongly correlated with perceived stress ([63]), suggesting that individual personality differences may shape moral reasoning under prolonged stress.

It is worth noting that most studies finding a positive correlation between stress and deontological decision-making were conducted pre-COVID (e.g., [130]), while studies supporting the opposite trend have emerged post-COVID ([131]). Although overall utilitarian tendencies showed no significant difference pre- and post-COVID (comparing data collected in 2014 and 2020), post-COVID participants were found to be less likely to make utilitarian decisions when personal rights were at stake ([5]). In other words, they were less inclined to compromise individual rights to achieve the greatest collective benefit—a trade-off included in many items within the CNI model. Given these changes in the utilitarian trend, it is plausible that the COVID-19 pandemic moderated the effect of stress on U-D decision-making.

#### 4.2.3. Other Observations

The current study found that women were less sensitive to consequences but more sensitive to norms and exhibited a greater preference for inaction, although the effects of Norm Sensitivity and Inaction Preference were marginal. These findings are consistent with prior research showing that women are generally less utilitarian ([6]) and exhibit higher Norm Sensitivity and Inaction Preference ([41]).

No significant effect of perceived stress on Inaction Preference was observed. Inaction Preference in the CNI model was introduced as a mechanism that is independent of moral norms or consequences. [41] ([41]) clarified that this dimension does not describe a stable psychological trait and may not be able to be generalized.

### 4.3. The Mediation Effect Between PSS and Norm Sensitivity Through PI Schema

The analysis revealed a significant mediation effect: a higher level of perceived stress is associated with lower Norm Sensitivity, and this relationship is partially mediated by the adoption of the Personal Interest schema. Perceived stress appears to heighten self-focused reasoning, leading individuals to prioritize personal interests, which, in turn, reduces their adherence to normative considerations when making moral judgments.

This finding aligns with the theoretical perspective that perceived stress can erode emotional responsiveness and cognitive engagement with broader societal norms. Stress may diminish the psychological salience of norms and ethical rules by increasing self-focus and reducing emotional bandwidth. As individuals under stress gravitate toward self-serving motivations, they may become less compelled to adhere to normative principles that emphasize the well-being and rights of others.

Moreover, this result supports the tentative explanation presented earlier: individuals experiencing heightened stress may develop emotional callousness or desensitization toward the implications of their actions on others. This emotional disengagement could “free” them from the moral impulse to prioritize deontological decision-making, which often involves adherence to norms, even at a personal cost.

While statistically significant, the mediation effect was modest in magnitude. Specifically, the inclusion of the PI schema as a mediator accounted for an additional 4.4% of the variance in Norm Sensitivity, suggesting that its practical impact may be limited.

The partial mediation observed in this study also underscores the complex interplay between stress, moral reasoning schemas, and decision-making processes. While PI schema adoption accounts for some of the stress-related declines in Norm Sensitivity, it is likely not the sole mechanism. Other factors, such as diminished cognitive flexibility, heightened emotional reactivity, and shifts in moral priorities under stress, may also contribute to this relationship. Future research should further explore these pathways to provide a more comprehensive understanding of how perceived stress reshapes normative decision-making tendencies.

### 4.4. Societal Changes and Generational Shifts—Another Potential Explanation of the Findings

The current findings align with these broader societal changes. With a predominantly young adult sample (mostly between 18 and 30 years old), this study found a positive correlation between perceived stress level and utilitarian decision-making, a positive correlation between stress and PI schema adoption, and a negative correlation between PI schema adoption and deontological decision-making. These results resonate with the observed generational shifts in stress levels and moral reasoning. 

Gen Z reports higher-than-average stress levels, surpassing Millennials at the same age, with stress levels rising steadily over the years ([3], [4]). Younger generations also exhibit more permissive attitudes toward utilitarian judgments ([6]; [57]; [82]), and longitudinal data from the Center for the Study of Ethical Development Reports indicate a steady increase in PI schema adoption and a decline in PC and MN schema adoption among younger individuals, suggesting a cultural shift in moral judgment and reasoning frameworks ([94]).

This generational shift is particularly evident in public reactions to morally complex cases such as the Killing of Brian Thompson. On 4 December 2024, the CEO of UnitedHealthcare, a major American health insurance company, was shot and killed in Midtown Manhattan, New York City ([123]). One key reason this seemingly indefensible murder sparked widespread discussion was the public perception that under Thompson’s leadership, the rate of medical claim approvals had significantly declined, exacerbating long-standing frustrations with the healthcare system. Consequently, some viewed the killing as understandable, if not justifiable. The Investor Dilemma and Tyrant Killing Dilemma employed in this study echo the moral complexity and societal discontent reflected in this case, where deliberate harm to a single individual—often portrayed as a morally corrupt figure in a position of power—is weighed against the potential to disrupt unjust systems and achieve broader societal benefits.

This approval perspective is particularly prevalent among younger generations. An [37] ([37]) revealed that while the majority of voters condemned the act, only 41% of voters aged 18–29 viewed it as unacceptable. This stands in stark contrast to the older generations, who overwhelmingly rejected this action. 

Thus, in addition to the discussions in the previous chapters, another potential explanation for the observed relationship between stress and moral judgment may not be causal but rather a byproduct of changing societal values. These include an increased emphasis on individual interests ([73]) and growing skepticism toward traditional institutions ([19]; [61]). Declining trust in systems perceived as unjust or ineffective may prompt younger individuals to challenge established norms and support actions they believe serve broader societal benefits, even at the expense of individual rights. This distrust in collective structures leads to a heightened focus on protecting individual interests ([116]). Furthermore, the erosion of trust in authority and systems undermines a sense of security, adding to the psychological burden on individuals ([19])—aptly illustrating Jean-Paul Sartre’s notion of “the heavy burden of freedom.”

### 4.5. Implications

The findings of this study have several important implications for moral education, stress research, and broader societal discussions on moral reasoning and decision-making.

#### 4.5.1. Informing Moral Education Programs

This study provides empirical support for integrating moral schemas and U-D decision-making tendencies into moral education frameworks. By identifying how moral schema adoption correlates with specific moral judgment processes, educators and policymakers can develop more targeted interventions. 

For example, educators should not only focus on fostering PC schema adoption but also on strategies to reduce PI schema adoption. The negative association between PI schema adoption and Norm Sensitivity suggests that individuals who strongly prioritize personal interests may be less inclined to adhere to established moral norms. Addressing this discrepancy in moral education could help counteract the decline in norm-based ethical reasoning and promote a more balanced moral framework—one that incorporates both universal ethical principles and sensitivity to social norms.

#### 4.5.2. Rising Stress Levels and Psychological Concerns

A concerning finding from this study is the elevated average perceived stress level among participants (20.9), which is significantly higher than the original normative scores (12–14) reported by [30] ([30]). This aligns with broader research indicating a steady rise in stress levels over the past two decades, particularly among young adults (e.g., 19.79 by [34], and 24.86 by [124], [124]). These findings underscore the urgent need to address stress as a critical factor influencing moral cognition and decision-making. 

The findings also indicate that higher stress levels are associated with increased PI schema adoption, greater reliance on utilitarian reasoning, and lower deontological judgments. This suggests that stress management interventions may play a crucial role in shaping moral development.

Recently, [80] ([80]) proposed that character education emphasizing virtues like *phronesis* contributes to adolescent mental health by providing a moral-existential framework for navigating ethical dilemmas and life challenges. *Phronesis* integrates the cognitive, emotional, and behavioral dimensions of morality, fostering the ability to navigate complex moral situations with sound judgment and habituated virtue. From an educational perspective, embedding *phronesis* within moral education can enhance students’ moral resilience, not only reinforcing moral reasoning but also cultivating the virtues necessary for moral action. Future moral education initiatives should integrate psychological resilience training with ethical and character education. Such an approach would not only support students’ mental health but also contribute to better moral reasoning and decision-making.

#### 4.5.3. Societal Trends and the Need for Moral Education

As discussed in the fourth point in the Section 4, the current findings align with broader societal changes, suggesting a generational shift in moral reasoning patterns—one that increasingly prioritizes personal interest and utilitarian moral judgment. These trends may be partially attributed to the growing erosion of trust in traditional institutions and in collective structures. As skepticism toward authority and systemic frameworks increases, individuals are increasingly burdened by moral decision-making in an uncertain social landscape. 

While such shifts may reflect evolving societal values, they also highlight the pressing need for renewed efforts in moral education. If younger generations are moving away from conventional moral schemas, educational interventions must adapt to ensure that ethical reasoning remains a well-supported and actively cultivated skill. By fostering critical moral reflection, reinforcing ethical responsibility, and addressing the psychological dimensions of decision-making, moral education can serve as a stabilizing force in an era of increasing moral complexity.

In summary, as stress levels rise and moral reasoning evolves, proactive educational initiatives can help individuals develop the cognitive, emotional, and ethical capacities necessary to navigate an ever-changing moral landscape. Further research should explore how moral education can be tailored to address these shifts, ensuring that ethical reasoning remains robust in the face of both psychological and societal challenges.

### 4.6. Limitations and Future Directions

This study has several limitations. First, the sample was relatively homogenous, consisting predominantly of female (84.6%) and Caucasian (84.9%) college students in the United States. This restricts the generalizability of our findings to broader and more diverse populations. Future research would benefit from replicating this study with participants from more diverse demographic and cultural backgrounds.

Additionally, the study relied exclusively on self-report measures. While widely used and validated, such measures are subject to social desirability bias and inaccurate self-assessment ([96]). Moreover, the PSS may not fully capture the multifaceted nature of stress ([88]). Future research should incorporate more objective measures of stress, such as biological markers (e.g., cortisol levels and heart rate variability) or ecological momentary assessments.

Third, the cross-sectional design prevents causal inferences between moral reasoning and the CNI dimensions. Longitudinal studies would help explore how these relationships evolve over time and whether changes in moral reasoning schemas predict shifts in the CNI dimensions.

Fourth, the data used in this study were collected prior to the manuscript preparation and may be somewhat dated. While the theoretical framework and analysis remain timely and relevant, we acknowledge that the age of the dataset may limit its applicability.

Finally, while this study examined the mediating role of the PI schema, other factors, like emotional regulation ([106]) and cognitive flexibility ([22]), may also influence moral judgment. Longitudinal or experimental designs could help disentangle these complex interactions and offer a more comprehensive view of the mechanisms linking stress to moral judgment.

Despite these limitations, the current study provides valuable insights into how perceived stress shapes moral reasoning and decision-making. By addressing these open questions and replicating the findings in diverse contexts, future research can further enhance our understanding of this relationship.

## Figures and Tables

**Figure 1 ejihpe-15-00106-f001:**
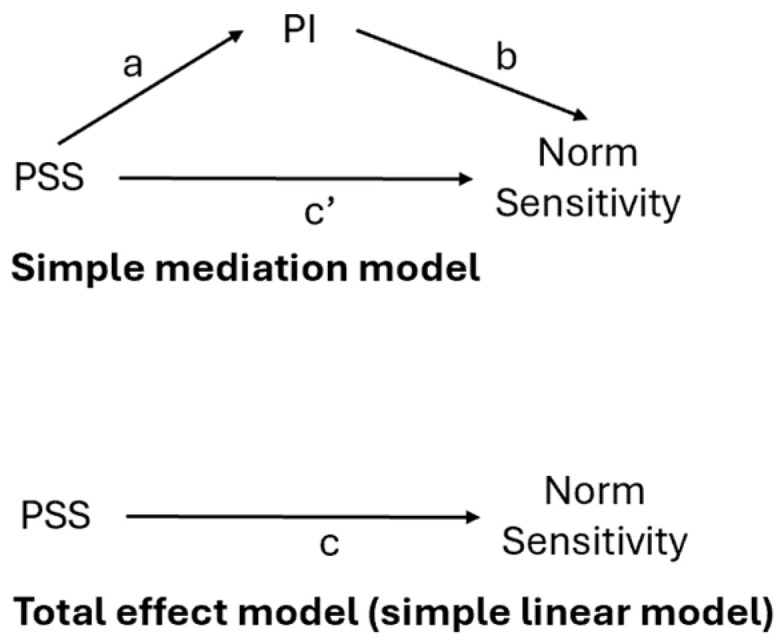
Mediation Model.

**Table 1 ejihpe-15-00106-t001:** Item effect check using data from [66] ([66]).

		Percentage of “Yes” Answers				Consequence Effect		Norm Effect	
		Prescriptive Norm		Proscriptive Norm					
Dilemma Name	Study No.	Benefits Greater Than Cost	Benefits Smaller Than Cost	Benefits Greater Than Cost	Benefits Smaller Than Cost	*F*	*p*	*F*	*p*
7. Dialysis	1a	84.47%	44.72%	54.04%	21.12%	80.04	<0.001	43.74	<0.001
	1b	82.49%	45.20%	53.67%	20.34%	82.33	<0.001	55.39	<0.001
8. Investor	1a	78.88%	64.60%	28.57%	14.29%	22.65	<0.001	112.91	<0.001
	1b	80.79%	67.23%	28.25%	13.56%	25.40	<0.001	168.71	<0.001
9. Tyrant Killing	1a	77.64%	52.80%	45.34%	23.60%	41.09	<0.001	38.33	<0.001
	1b	74.01%	62.15%	46.33%	26.55%	23.94	<0.001	50.07	<0.001
10. Rwanda	1a	88.20%	82.61%	26.09%	16.15%	9.84	0.002	235.45	<0.001
	1b	86.44%	83.62%	30.51%	20.34%	7.97	0.005	233.65	<0.001
11. Mercy Killing	1a	88.20%	61.49%	40.99%	9.94%	65.47	<0.001	196.32	<0.001
	1b	83.05%	60.45%	40.11%	16.38%	48.45	<0.001	132.48	<0.001
12. Nazi Occupation	1a	88.82%	80.12%	14.91%	13.04%	5.40	0.021	339.26	<0.001
	1b	83.62%	76.84%	20.90%	15.25%	7.37	0.007	263.63	<0.001

**Table 2 ejihpe-15-00106-t002:** Descriptive statistics and correlations between variables.

Variable		Mean	*SD*	*n*	1	2	3	4	5	6	7
Moral Reasoning Schema	1. PI	22.98%	0.16	337	1						
	2. MN	24.80%	0.18	337	−0.340 **	1					
	3. PC	52.21%	0.20	337	−0.502 **	−0.643 **	1				
Moral Dilemma Judgment	4. C	−0.80	0.59	337	0.09	−0.206 **	0.116 *	1			
	5. N	0.40	0.67	337	−0.229 **	0.201 **	0.002	−0.367 **	1		
	6. I	0.40	0.56	337	0.041	−0.049	0.012	0.152 **	−0.317 **	1	
Stress	7. PSS	20.91	6.96	337	0.107 *	−0.053	0.039	0.116 *	−0.188 **	−0.041	1

** Correlation is significant at the 0.01 level (2-tailed). * Correlation is significant at the 0.05 level (2-tailed).

## Data Availability

The original data presented in the study are openly available in the Open Science Framework at https://osf.io/bp2m6/ (accessed on 6 April 2025).

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
