# Peer review of "Perceived Stress and Society-Wide Moral Judgment"

_ejihpe, 2025, doi:10.3390/ejihpe15060106_

Round 1
Reviewer 1 Report
Comments and Suggestions for Authors
The manuscript offers a fresh and well-executed look at how perceived stress relates to moral judgment, using two theoretical frameworks: the neo-Kohlbergian approach and the CNI model (operationalised through the EIRTree model). It stands as a replication of Zhang et al.’s (2018) study using the CNI model, but now extending it into other cultures by using a US sample. Besides, it has the novelty of studying moral decisions under stress using the neo-Kohlbergian approach. The study is solid both methodologically and theoretically, and it brings meaningful insights into how stress might influence moral thinking and reasoning. I found the way it combines moral schema theory with process-level modeling through the CNI framework especially interesting. It helps bridge developmental and decision-making perspectives on moral judgment.
A few highlights that stand out:
- The authors do a good job bringing together the neo-Kohlbergian framework and the CNI model. It’s a thoughtful combination that allows them to explore different but complementary aspects of moral decision-making.
- The study uses well-validated tools (bDIT, PSS-10, CNI test), and the data cleaning and statistical analyses are thorough, including mediation models and diagnostic checks.
- The sample size (N = 337) is appropriate, and the authors clearly explain their sampling decisions with solid power calculations using G*Power. That adds confidence in the robustness of the findings.
- Using gender as a controlling variable since women are overrepresented in the study.
- The study fills a clear gap in the literature by connecting perceived stress with moral schemas and judgment patterns. The replication of earlier CNI findings with a U.S. sample is a nice bonus and adds further value.
That said, I think the manuscript could benefit from a few clarifications and refinements. I would suggest the following areas for improvement:
The introduction lays out the frameworks well, but it would help to more clearly explain why the authors expected increased Consequence Sensitivity under stress, especially given the inconsistency of previous studies regarding this issue. Framing this in terms of possible dual-process theory conflicts, or other perspectives (like cognitive load or stress-related utilitarian thinking), could strengthen the logic.
In the mediation analysis, the indirect effect is statistically significant but quite small. While that’s worth noting, it might not be practically meaningful. It would help if the authors reported effect sizes more explicitly (e.g. using standardized coefficients or R² changes would add clarity).
Still in the frame of the statistical approach, the quadratic analyses (like the potential curvilinear link between stress and PI schema) didn’t improve model fit. These parts could be shortened or flagged as exploratory to avoid overinterpreting them.
In terms of interpreting the findings, some findings (like the unexpected positive correlation between stress and Consequence Sensitivity) might benefit from deeper cross-cultural reflection. The manuscript briefly mentions cultural differences (e.g., China vs. USA), but doesn’t dig into how these might shape the stress–morality connection.
Finally, the limitations section could be more upfront about the reliance on self-report measures and the fairly homogenous sample (mostly female college students), as these factors may limit broader generalizability.
Author Response
Reviewer 1
The manuscript offers a fresh and well-executed look at how perceived stress relates to moral judgment, using two theoretical frameworks: the neo-Kohlbergian approach and the CNI model (operationalised through the EIRTree model). It stands as a replication of Zhang et al.’s (2018) study using the CNI model, but now extending it into other cultures by using a US sample. Besides, it has the novelty of studying moral decisions under stress using the neo-Kohlbergian approach. The study is solid both methodologically and theoretically, and it brings meaningful insights into how stress might influence moral thinking and reasoning. I found the way it combines moral schema theory with process-level modeling through the CNI framework especially interesting. It helps bridge developmental and decision-making perspectives on moral judgment.
A few highlights that stand out:
- The authors do a good job bringing together the neo-Kohlbergian framework and the CNI model. It’s a thoughtful combination that allows them to explore different but complementary aspects of moral decision-making.
- The study uses well-validated tools (bDIT, PSS-10, CNI test), and the data cleaning and statistical analyses are thorough, including mediation models and diagnostic checks.
- The sample size (N = 337) is appropriate, and the authors clearly explain their sampling decisions with solid power calculations using G*Power. That adds confidence in the robustness of the findings.
- Using gender as a controlling variable since women are overrepresented in the study.
- The study fills a clear gap in the literature by connecting perceived stress with moral schemas and judgment patterns. The replication of earlier CNI findings with a U.S. sample is a nice bonus and adds further value.
That said, I think the manuscript could benefit from a few clarifications and refinements. I would suggest the following areas for improvement:
The introduction lays out the frameworks well, but it would help to more clearly explain why the authors expected increased Consequence Sensitivity under stress, especially given the inconsistency of previous studies regarding this issue. Framing this in terms of possible dual-process theory conflicts, or other perspectives (like cognitive load or stress-related utilitarian thinking), could strengthen the logic.
Response: Thank you for this thoughtful comment. We appreciate the suggestion to clarify the theoretical grounding regarding stress and moral judgment. We would like to clarify that our original hypothesis did not propose an increase in Consequence Sensitivity under stress. Rather, as stated around line 235, we hypothesized that higher levels of perceived stress would be associated with increased Norm Sensitivity and/or decreased Consequence Sensitivity, in line with prior research indicating a general shift toward deontological reasoning under stress (e.g., Li et al., 2019; Starcke et al., 2012).
That said, we agree that the introduction could benefit from strengthening the theoretical background. In the revised manuscript, we have added a relevant section to more explicitly situate our hypotheses within dual-process theory and cognitive load perspectives, emphasizing how stress may impair deliberative processing and thereby reduce Consequence Sensitivity, around line 247. The revised paragraph now reads:
This expectation also aligns with the dual-process theory (Greene, 2014), which proposes that moral decisions result from the interplay between intuitive, emotion-driven processes (System 1) and deliberative, reasoning-based processes (System 2). Under stress, the capacity for deliberative reasoning (System 2) is often compromised, leading individuals to rely more heavily on intuitive processes (System 1). Consequently, higher perceived stress levels are anticipated to correlate with increased Norm Sensitivity or decreased Consequence Sensitivity, reflecting a shift toward deontological reasoning.
In the mediation analysis, the indirect effect is statistically significant but quite small. While that’s worth noting, it might not be practically meaningful. It would help if the authors reported effect sizes more explicitly (e.g. using standardized coefficients or R² changes would add clarity).
Response: We thank the reviewer for the helpful suggestion regarding effect size reporting. In response, we now report the R² change associated with the inclusion of the mediator. Specifically, the explained variance in the outcome variable increased from R² = .0353 (total effect model) to R² = .0793 (full mediation model), reflecting an R² change of .044. This has been added to the Results section for clarity (line 576). We also highlighted the limited practical impact of this mediation effect in the Discussion section (line 590 & 757).
Still in the frame of the statistical approach, the quadratic analyses (like the potential curvilinear link between stress and PI schema) didn’t improve model fit. These parts could be shortened or flagged as exploratory to avoid overinterpreting them.
Response: The quadratic analyses are flagged as exploratory as seen in the title under Method “2.4.3. Exploring Quadratic Relationship” (line 456). We also tried to avoid overinterpreting them by not mentioning it in the Discussion section.
In terms of interpreting the findings, some findings (like the unexpected positive correlation between stress and Consequence Sensitivity) might benefit from deeper cross-cultural reflection. The manuscript briefly mentions cultural differences (e.g., China vs. USA), but doesn’t dig into how these might shape the stress–morality connection.
Response: Thank you for this thoughtful suggestion. We did consider elaborating further on potential cultural differences, particularly in relation to the divergent findings between Zhang et al. (2018), conducted in China, and the present study, conducted in the United States. However, it is worth noting that Zhang et al.’s results are largely consistent with a broader body of literature—spanning both Western and non-Western contexts—reporting an association between stress and deontological tendencies. Our contrary finding, in contrast, stands out not only from Zhang et al. (2018)’s study, but also from most existing research.
Given this broader divergence, we believe that the unexpected result in our study is less likely to be explained primarily by cultural differences, and more likely reflects other mechanisms discussed in the manuscript. To avoid overinterpreting or misleading readers about the role of culture in this particular finding, we chose not to foreground cultural explanations.
Finally, the limitations section could be more upfront about the reliance on self-report measures and the fairly homogenous sample (mostly female college students), as these factors may limit broader generalizability.
Response: Thank you for this helpful comment. In response, we have revised the Limitations section to more explicitly acknowledge both the reliance on self-report measures and the demographic homogeneity of our sample (line 872). We now discuss how these factors may limit the generalizability of our findings and introduce additional recommendations for incorporating objective stress assessments in future research. We appreciate the reviewer’s attention to these important methodological considerations.
Reviewer 2 Report
Comments and Suggestions for Authors
- What is the main question addressed by the research?
- This study examines the relationship between perceived stress and society-wide moral judgement by integrating two influential frameworks.
- Do you consider the topic original or relevant to the field? Does it address a specific gap in the field? Please also explain why this is/ is not the case.
- It is relevant to the field. However, data is outdated. Please consider to do the data collection this year. It address the literature gap.
- What does it add to the subject area compared with other published material?
- Provide empirical evidence.
- What specific improvements should the authors consider regarding the methodology?
- This is a cross sectional study, need to control common method bias. Now descriptive result has been presented. Consider to use structural equation modeling.
- Are the conclusions consistent with the evidence and arguments presented and do they address the main question posed? Please also explain why this is/is not the case.
- Yes. Need to add managerial implications and theoretical contributions.
- Are the references appropriate?
- Yes, they are.
- Any additional comments on the tables and figures.
- Nil.
Author Response
Reviewer 2
- What is the main question addressed by the research?
- This study examines the relationship between perceived stress and society-wide moral judgement by integrating two influential frameworks.
- Do you consider the topic original or relevant to the field? Does it address a specific gap in the field? Please also explain why this is/ is not the case.
- It is relevant to the field. However, data is outdated. Please consider to do the data collection this year. It address the literature gap.
Response: Thanks for your advice and suggestion.
To address the concern, we noted in the limitation section that the data is outdated at the time of submission (line 895).
The current study was conducted as a part of the first author’s dissertation research, so it took a while until the manuscript was ready for submission by passing the dissertation defense. At this point, the first author graduated from the former institute, so it is practically difficult to collect additional data.
- What does it add to the subject area compared with other published material?
- Provide empirical evidence.
- What specific improvements should the authors consider regarding the methodology?
- This is a cross sectional study, need to control common method bias. Now descriptive result has been presented. Consider to use structural equation modeling.
Response: Thank you for the valuable suggestion. We agree that SEM would be a useful tool to address measurement errors and biases while analyzing the relationships among variables of interest. Despite the potential benefit, we assume that SEM might not be an ideal analysis method within the context of the current study, so we decided to employ other analysis methods, such as correlational and regression analysis. One point to consider is that two main variables of interest, the bDIT and CNI scores, are not based on latent factor structures. Instead, they are more like test performance scores, so those scores per se can be and shall be treated as observed, not latent, variables. Such a fact may suggest that SEM might not be suitable to be conducted in the present study.
- Are the conclusions consistent with the evidence and arguments presented and do they address the main question posed? Please also explain why this is/is not the case.
- Yes. Need to add managerial implications and theoretical contributions.
Response: Thanks for the valuable suggestion. We have now added an Implications section that addresses three key areas: (1) how the study could inform moral education, (2) the raising stress level and psychological concerns, and (3) broader societal trends—such as increased stress, growing approval of utilitarian reasoning, and a decline in the adoption of high-level moral schemas—which together highlight the need for a more adaptive approach to moral education (line 895).
- Are the references appropriate?
- Yes, they are.
- Any additional comments on the tables and figures.
- Nil.
Reviewer 3 Report
Comments and Suggestions for Authors
The study is properly organized and referenced. It miss just some sensitivity regard the high presence of cultural and confusing variable beyod the simple stress withi decision process
Author Response
Reviewer 3
The study is properly organized and referenced. It miss just some sensitivity regard the high presence of cultural and confusing variable beyod the simple stress withi decision process
Response: Thank you for this thoughtful suggestion. We did consider elaborating further on potential cultural differences, particularly in relation to the divergent findings between Zhang et al. (2018), conducted in China, and the present study, conducted in the United States. However, it is worth noting that Zhang et al.’s results are largely consistent with a broader body of literature—spanning both Western and non-Western contexts—reporting an association between stress and deontological tendencies. Our contrary finding, in contrast, stands out not only from Zhang et al. (2018)’s study, but also from most existing research.
Given this broader divergence, we believe that the unexpected result in our study is less likely to be explained primarily by cultural differences, and more likely reflects other mechanisms discussed in the manuscript. To avoid overinterpreting or misleading readers about the role of culture in this particular finding, we chose not to foreground cultural explanations.
Round 2
Reviewer 2 Report
Comments and Suggestions for Authors
I sent through all the authors' responses. It is a pity that almost all my concerns cannot be addressed. I read the revised paper, and the implications and limitations have been improved. The literature are outdated. need to revisit. No theoretical foundation and theoretical contribution. Please do not use footnote format. Consider recollect the samples using employees. Why US samples are used? Is it representative for all the people?
Author Response
Thanks for your comments on the revised manuscript. In the current version, we highlighted parts that were changed from the original version for your convenience.
Comment 1. The literature are outdated. need to revisit.
Response 1. Thank you for your comment about the literature cited in the manuscript. Although some previous works referred were relatively old (e.g., those about the Neo-Kohlbergian theory), the majority of the literature that we cited to establish the foundation of the current work, particularly that about the CNI model, was from works done since less than a decade ago (e.g., Gawronski et al., 2017, 2018). Furthermore, additional works on stress and moral psychology were published in 2012 or thereafter (in fact, many of them were published in 2020 or thereafter), we believe that our literature review was based on the most up-to-date research in the field except for when introducing the classical theories (e.g., the Neo-Kohlbergian model and the dual process model) was needed.
Comment 2. No theoretical foundation and theoretical contribution.
Response 2. We appreciate your comment on the theoretical foundation and contribution. In fact, as you can see from the introduction, our study was firmly founded on three theoretical frameworks in the field, the Neo-Kohlbergian model (for the bDIT), CNI model (for the CNI), and research on stress and decision-making. Also, in the discussion section, as you can find, we discussed further implications of the current study on moral education and mental wellbeing. Given the EJIHPE's main purpose is to contribute to... "education in a coherent and practical way in order to study the human being from different perspectives or in different contexts (https://www.mdpi.com/journal/ejihpe/about)," we assume that the current paper with in-depth discussion on practical implications will be able to make significant contributions to the journal as stated in its aims and scope.
Comment 3. Please do not use footnote format.
Response 3. Thanks a lot for your comment on the formatting. We moved the contents in the footnote to the main text in the revised manuscript.
Comment 4. Consider recollect the samples using employees.
Response 4. We appreciate your suggestion on the additional sample. Although we agree with you that employing diverse samples may be beneficial in principle, we decided to focus on participants attending a school since the main foci of the current study were moral development and education, particularly those among adolescents within educational settings. Also, given the EJIHPE is a journal concerned about education, using student samples may be consistent with the journal's purpose in general.
Comment 5. Why US samples are used? Is it representative for all the people?
Response 5. Thanks for your comment about the samples. As mentioned in the revised manuscript, given the current study was based on a PhD dissertation written by a student who conducted the current study in the US, the participants were recruited in the US. We agree with you that such may significantly limit the potential generalizability of the current study, we acknowledged the limitation in the limitations section. In a long run, additional studies shall be done across different countries and cultures to be able to examine whether the current findings are replicable.